# Delayed antibiotic prescribing for respiratory tract infections: protocol of an individual patient data meta-analysis

Beth Stuart,[1] Hilda Hounkpatin,[1] Taeko Becque,[1] Guiqing Yao,[2] Shihua Zhu,[1] Pablo Alonso-Coello,[3] Attila Altiner,[4] Bruce Arroll,[5] Dankmar Böhning,[6] Jennifer Bostock,[7] Heiner C C Bucher,[8] Mariam de la Poza,[9] Nick A Francis,[10] David Gillespie,[11] Alastair D Hay,[12] Timothy Kenealy,[5] Christin Löffler,[13] Gemma Mas-Dalmau,[14] Laura Muñoz,[15] Kirsty Samuel,[16] Michael Moore,[1] Paul Little[1]

BS and HH contributed equally.

For numbered affiliations see end of article.

**Correspondence to**
Dr Beth Stuart;
bls1@soton.ac.uk

## ABSTRACT

**Introduction** Delayed prescribing can be a useful strategy to reduce antibiotic prescribing, but it is not clear for whom delayed prescribing might be effective. This protocol outlines an individual patient data (IPD) meta-analysis of randomised controlled trials (RCTs) and observational cohort studies to explore the overall effect of delayed prescribing and identify key patient characteristics that are associated with efficacy of delayed prescribing.

**Methods and analysis** A systematic search of the databases Cochrane Central Register of Controlled Trials, Ovid MEDLINE, Ovid Embase, EBSCO CINAHL Plus and Web of Science was conducted to identify relevant studies from inception to October 2017. Outcomes of interest include duration of illness, severity of illness, complication, reconsultation and patient satisfaction. Study authors of eligible papers will be contacted and invited to contribute raw IPD data. IPD data will be checked against published data, harmonised and aggregated to create one large IPD database. Multilevel regression will be performed to explore interaction effects between treatment allocation and patient characteristics. The economic evaluation will be conducted based on IPD from the combined trial and observational studies to estimate the differences in costs and effectiveness for delayed prescribing compared with normal practice. A decision model will be developed to assess potential savings and cost-effectiveness in terms of reduced antibiotic usage of delayed prescribing and quality-adjusted life years.

**Ethics and dissemination** Ethical approval was obtained from the University of Southampton Faculty of Medicine Research Ethics Committee (Reference number: 30068). Findings of this study will be published in peer-reviewed academic journals as well as General Practice trade journals and will be presented at national and international conferences. The results will have important public health implications, shaping the way in which antibiotics are prescribed in the future and to whom delayed prescriptions are issued.

**PROSPERO registration number** CRD42018079400.

## Strengths and limitations of this study

► This study uses individual patient data (IPD) from randomised controlled trials and observational studies to investigate the clinical effectiveness and potential cost-savings of a delayed antibiotic prescribing, both overall and for key subgroups of people.

► IPD provides sufficient statistical power to explore interactions between treatment groups and patient characteristics.

► The IPD will only include data from studies for which the data are available.

► Observational studies will use propensity score approach to adjust for confounding by indication on measured covariates.

► Findings from this study may inform GPs decisions on prescribing and increase use of appropriate delayed prescribing which in turn can help to reduce antibiotic consumption and antimicrobial resistance.

## INTRODUCTION

Antimicrobial resistance is a cause for great concern prompting calls for action at the local, national and international level to prevent 'overuse, misuse and abuse' of antibiotics,[1] particularly in primary care where antibiotics are most prescribed.[2] About 60% of antibiotics prescribed in primary care are for respiratory tract infections (RTIs).[3] However, most infections are self-limiting, symptomatic benefit from antibiotics is modest[4–6] and patients prescribed antibiotics for RTIs are more likely to carry antibiotic resistant commensal bacteria and develop resistant infections.[7]

The Standing Medical Advisory Committee report recommends that the fewest number of antibiotic courses should be prescribed for the shortest period possible.[8] However, in a

primary care context, it can be difficult to tell whether antibiotics are appropriate for an individual patient. It is not always clear which patients are at risk of prolonged illness or developing complications nor whether the RTI is bacterial or viral in nature. Point of care tests such as C reactive protein and clinical scores both show promise in helping to guide GP management decisions.[9–12] Delayed prescribing is a useful strategy that may be used on its own or in conjunction with clinical scores and point of care testing. It allows the patient to collect a prescription to be taken if their symptoms do not start to improve within a specific duration of their initial consultation, providing a potential aid to negotiating treatment acceptable to the patient and provider while reducing inappropriate prescribing.[13 14] Delayed prescribing is recognised as part of the toolkit available to GPs to help reduce antibiotic use, especially in the context of consultations where patients expect to receive an antibiotic prescription and is part of the National Institute for Health and Care Excellence guidelines.[3] A recent large prospective primary care cohort (MRC DESCARTE) demonstrated that delayed prescription was likely to be as effective as immediate antibiotic in reducing complications and more effective at reducing reconsultations.[4]

While in some situations delayed antibiotic prescribing is appropriate, for other patients it may be unsuitable and controversial.[15] It is important to understand which subgroups of patients may require immediate antibiotics to avoid complications and which patients might benefit from a delayed or no prescribing strategy.[4] A 2013 Cochrane review[16] of 10 trials found that a delayed prescribing strategy was not significantly different to immediate prescribing in terms of clinical outcomes for cough and cold. In patients with acute otitis media and sore throat immediate antibiotics were more effective than delayed for reducing fever, pain and malaise in some studies. However, the review noted a high level of heterogeneity between studies made combining them in a traditional meta-analysis difficult and did not allow sufficient power for the examination of subgroups or complications either in meta-analysis or in meta-regression.

Where aggregation is possible, a traditional meta-analysis or even meta-regression can be used to determine the overall main effect of delayed antibiotic prescription. Although such techniques can also be used to explore how effect varies across study characteristics, these techniques lack statistical power and are prone to confounding as patient-level characteristics are not taken into account.[17] An individual patient data (IPD) meta-analysis is considered the gold standard for meta-analyses and can overcome the limitations of meta-analysis of aggregate data. In an IPD meta-analysis, original data for all participants in each study are obtained and synthesised. This allows sufficient statistical power[18] to explore interactions between treatment groups and patient characteristics. This would allow us to identify key groups in which delayed prescribing or no prescribing might be inadvisable due to longer durations of illness, greater severity of illness

or higher risk of complications/reconsultation. These findings may inform GPs decisions on prescribing and increase use of appropriate delayed prescribing which in turn can help to reduce antibiotic consumption and antimicrobial resistance.

This protocol describes an IPD meta-analysis of both randomised controlled trials (RCTs) and observational cohort studies of delayed antibiotic prescribing for acute respiratory infections. Although RCTs represent the 'gold standard' design for assessing the impact of an intervention, the participants in these studies may differ systematically from the patients encountered in everyday clinical practice due to the inclusion criteria. By including observational studies in a meta-analysis, with suitable techniques to control for potential confounding by indication, it may be possible to obtain better estimates of the effect of delayed prescribing in routine practice.

## METHODS
### Aims and objectives
This IPD study will investigate the clinical effectiveness and potential cost-savings of a delayed antibiotic prescribing approach, both overall and for key subgroups of patients. The study started in October 2017 and is due to be completed in September 2020.

The objectives of this study are as follows:
► To achieve more accurate estimates of the clinical effectiveness of a delayed prescribing strategy on the primary outcome, symptom severity, by harmonising the outcomes from all trials and observational studies which have included this strategy.
► To estimate the clinical effectiveness of delayed prescribing for secondary outcomes: duration of illness, development of complications, reconsultation and patient satisfaction.
► To explore whether there are key subgroups (informed by the literature and described below) for whom delayed antibiotic prescribing may or may not be beneficial. Subgroups will be: prior duration of illness (above/below median for the condition), age (under 16 years, 16–64 years, age over 65 years), fever at baseline consultation (>37.5°C), comorbid conditions including lung comorbidity such as asthma or chronic obstructive pulmonary disease (COPD) and severity of symptoms at baseline consultation. The effectiveness of delayed antibiotic prescribing for subgroups of symptom complexes such as cough (acute bronchitis), pharyngitis or otitis media will also be explored.
► To investigate whether symptom trajectories are influenced by antibiotic prescribing/consumption.
► To estimate the costs of treatment of patients with RTIs (from the UK National Health Service (NHS) and Personal Social Services (PSS) perspective data) and investigate any potential cost-savings and cost-effectiveness in terms of antibiotic usage of delayed antibiotic prescriptions compared with antibiotic use.

► To identify priorities for future delayed prescribing research.

## Study approach

A full systematic review will be conducted to identify and select eligible papers. Study authors of eligible papers will be contacted and invited to contribute raw data.

## Systematic review to identify eligible papers
### Eligibility criteria

*Design:* RCTs or eligible observational cohort studies.

*Population:* all patients attending primary, ambulatory or acute care settings with a RTI.

*Intervention:* delayed antibiotic prescription.

*Comparator:* immediate antibiotic prescription or no antibiotic prescription.

*Outcomes:* severity of illness (symptom severity), duration of illness, complication, reconsultation, patient satisfaction, costs and quality of life.

Studies on antibiotic prescribing that are not an RCT or observational cohort (eg, survey or cross-sectional studies and case–control studies) will be excluded. Studies of hospital inpatients are also excluded.

## Search strategy for identification of studies

A systematic search of the databases Cochrane Central Register of Controlled Trials, Ovid MEDLINE, Ovid Embase, EBSCO CINAHL Plus and Web of Science will be conducted to identify relevant papers published from inception to October 2017. The search strategy will be based on criteria set out by the Cochrane Collaboration in a recently published systematic review and is available in the online supplementary appendix.[19] The International Standard Randomised Controlled Trial Number Registry, a primary clinical trial registry recognised by WHO and ICMJE that accepts and records all clinical research studies in order to improve the publicly available information about clinical studies, will also be searched for any relevant trials. Literature search results will be exported to EndNote and then uploaded to COVIDENCE, a web-based software that facilitates collaboration among reviewers during the study selection process. Full-text articles will be uploaded to COVIDENCE. Two reviewers (HH and TB) will independently assess articles to determine eligibility for inclusion and any discrepancies will be resolved by discussion or referred to a third reviewer (BS). Reference lists of included articles will be reviewed to identify other potential papers not retrieved in the initial search. Contributing authors and content experts will be asked if they have, or are aware of, any additional studies (published or unpublished).

## Risk of bias assessment and certainty of evidence assessment

Two reviewers will independently assess the risk of bias of each included study. RCTs will be assessed using the Cochrane risk of bias tool that includes items on allocation bias (random sequence generation, allocation concealment, baseline imbalance), departures from

intended interventions (participant and study personnel blinding, deviations from intended interventions and analysis in groups to which they were randomised), attrition bias and appropriate methods to account for missing data, detection bias (blinding of outcome assessors) and selective outcome reporting.[20 21] The "Risk Of Bias In Non-randomised Studies - of Interventions" (ROBINS-I) tool will be used to assess quality of each observational study. The ROBINS-I tool contains items on bias due to confounding, selection bias, bias due to deviations from intended intervention, bias due to missing data and selective reporting.[22] We will use GRADE to rate the overall certainty (quality) of evidence that includes the evaluation of risk of bias, inconsistency, indirectness, imprecision and publication factors.[23]

## Data extraction and database creation

Authors of eligible trials and observational studies will be contacted via an email, or where this is not possible by a letter, outlining the study aims. They will be invited to collaborate and share their data in a format of their choice. A data sharing agreement may be provided on request. A complete database (containing data on all available measures) rather than key variables used in the publication will be requested. If resource usage and quality of life data were collected in separate forms, such information should be provided too. This will allow us to recalculate the primary outcomes where necessary to ensure a consistent approach has been used across studies with the aim of reducing the level of between-study heterogeneity. Collaborators will be able to provide the data in a format of their choice. Study data will be considered unavailable in the event that none of the authors have responded to multiple contact attempts or if all contacted study authors indicate they no longer have access to the data.

Data will be checked by comparing key variables (eg, size of sample, descriptive statistics of demographic and outcome measures) with published data. We will aim to reproduce individual study results to ensure the current analysis is consistent with what has been done in each study. Collaborators will be contacted if important discrepancies are identified and asked for clarification. The level of missing data within each included study will be assessed and discussed with study collaborators. If less than 5% of data is found to be missing on key baseline characteristics and outcome measures, the data will be analysed on a complete cases basis. However, if there is substantial missing data, the pattern and nature of the missingness will be explored and multiple imputation by chained equations will be used if appropriate.

A copy of the data to be included in the IPD meta-analysis will be converted for analysis in STATA V.14 or above. A database will be created containing study ID, patient ID, outcomes measures, subgroups of interest, antibiotic prescribing approach and propensity scores, where necessary. Outcome measures from the different studies will be harmonised once data from all study authors have been received. In the case that coding for an outcome in a

study differs greatly from other studies, it may be possible to simplify measures to allow the study to contribute data or exclude the study from certain analyses. Any difficulties with harmonisation will be discussed with the full collaborators group to decide on the most meaningful approach. Harmonised data from the different studies will be aggregated (using STATA) into a single large IPD database, with an indicator variable to identify patients from the same trial. The IPD database will be rechecked for accuracy.

The key resource information on medication, primary care consultation, hospital inpatients, hospital day cases, outpatient visits, accident and emergency (A&E) attendance from individual studies will be recorded if such information were collected and length of usage and frequency of attendance will be adjusted to the same time periods. We will use the UK national published tariff (British National Formulary (BNF), national reference cost and PSS Research Unit (PSSRU)) to cost-associated resource usage.

## Analysis

Study and patient level characteristics will be described for all studies that contribute IPD. Characteristics of any studies that declined or were unable to provide data will be considered in order to determine the extent to which the included studies are a representative sample. A traditional meta-analysis may be performed, using data from published papers, to test for differences between studies included in the IPD meta-analysis and those that could not provide data.[24]

Heterogeneity of eligible studies will be summarised using an $I^2$ statistic (tested by Higgins $I^2$ test). A substantial statistical heterogeneity will be considered if $I^2$ statistic is >50%.[21] Sources of heterogeneity will be explored and reported and a random effects model will be used for all analyses.

### IPD meta-analysis

IPD meta-analysis will be conducted using a one-stage approach. The one-stage approach combines all the data in a single meta-analysis based on a regression model stratified by study.[25] One-stage analysis is often more appropriate for exploring treatment-covariate interactions as it has increased power and is less likely to suffer from aggregation bias.[26] The aggregated IPD will be examined using multilevel regressions clustering on the individual study level to take into account any heterogeneity between studies. Using the aggregated IPD dataset, we will calculate the key outcome measures on a consistent basis. The model selected will be one that is appropriate to the outcome measure of interest. For example, assuming the underlying assumptions of the model are met, a linear regression will be used to model the severity of symptoms, a proportional-hazards model will be used to assess the duration of illness, or a suitable count model where time to event data is not available, and logistic regression models will be used for the complications/

reconsultation outcome, side effects and patient satisfaction. All models will control for baseline severity of illness and all individuals will be included on a modified intention to treat (ITT) basis (ie, as randomised).[27]

To explore whether there are differences in treatment effects for certain sub-groups, we will consider five prespecified subgroups which have been identified as potential effect modifiers by previous research: prior duration of illness (above/below median for the condition), age (under 16 years, 16–64 years, age over 65 years), fever at baseline consultation (>37.5°C), comorbid condition including lung comorbidity such as asthma or COPD, and severity of symptoms at baseline consultation. We will also explore differences according to the diagnostic group (acute sore throat, cough/chest infection, otalgia/otitis media, upper RTI). The analyses described above will be repeated, testing for interactions between treatment allocation and sub-group characteristic.

Symptom trajectories in patients who do not take antibiotics will also be explored. For symptoms recorded in symptom diaries, we will calculate the median and IQR of the reported time until all symptoms have settled completely, in those patients who did not receive an antibiotic prescription or who received a delayed prescription but did not fill it (where these data are available). As sensitivity analyses, we will also explore duration of moderately severe of symptoms. It is possible that the use of multiple outcomes (symptom severity, development of complications, patient satisfaction) to determine clinical effectiveness may result in varying conclusions. We will report the results for each outcome, though the discussion and recommendations will be based primarily on the primary outcome– symptom severity.

Work on cross design synthesis by the Program and Methodology Division of the US General Accounting Office (1992) has provided a theoretical framework for combining database data and RCT data. Drawing on this work, the current review will apply a similar approach to the synthesis of observational cohort and RCT data.[28] Synthesis of observational studies will additionally use inverse probability weighting by propensity score analysis to adjust for confounding by indication on measured covariates. This serves to balance observational studies on key covariates) so that the distribution of baseline covariates is similar between treated and untreated subjects, making observational studies more like those in a RCT.[29 30] Potential covariates to be included in the propensity score include: demographic factors (eg, age, sex), comorbid health conditions and signs and symptoms at baseline consultation. Observational studies after this adjustment for potential confounding variables will be included in the main analysis if appropriate. However, sensitivity analyses will explore whether the estimates from observational cohort studies differ from those found in the RCTs. If there is evidence of substantial differences, data from observational studies and RCTs will not be pooled in a single meta-analysis, and reasons for these differences will be explored narratively.

## Economic analysis

The economic evaluation will be taken from the NHS and PSS perspective, as most of the studies in this area have collected such data. The NHS and PSS perspective is the most commonly used approach to economic evaluations conducted in the UK. It covers medication, primary care consultation, walking in centre, NHS telephone service, outpatient attendance, A&E visit and hospital admission. Outcomes will be differences in cost, costs per antibiotic prescription avoided and costs per quality of life gained. Individual patient resource usage data such as medication use, GP consultations, outpatient visits, A&E attendance and use of secondary care service from each individual trial or observational study will be costed using UK published data sources (BNF, national reference costs and PSSRU). Total costs for individual patients during study periods will be calculated. We will also take a societal perspective by including the societal cost of antimicrobial resistance and time of work due to illness in our modelling and sensitivity analyses.

A decision analytical model will be developed to project the potential saving (due to reduced antibiotic usage, and reduced frequency of attending primary care consultation) and cost-effectiveness in terms of reduced antibiotic usage, and quality-adjusted life years (QALY) gained of delayed antibiotics. The model will consider reinfection and present the results at different time lengths, up to a maximum of 5 years. The model will be an individual-based micro-simulation taking into account individual risk profiles with associated risk equations. The model will simulate individual treatment pathways with defined baseline characteristics and will capture the time spent at different levels of disease severity, time to recovery and recurrence rates. The probabilities of staying or transitioning to another state will be derived from the statistical analyses. The clinical analyses based on all data will provide more robust estimates to populate the input values for the data. The costs and cost-effectiveness of subgroups will be explored in the decision model.

Quality of life questionnaire will be translated into utility scores based on the UK tariff appropriate to the instruments used in each study. The different instruments (such as EQ5D, SF6D or Health Utilities Index (HUI)) will be recorded and weighted in the aggregated analyses. Utility scores at each measurement point will be estimated, and a mean utility score for symptom severity will be calculated. For studies where no quality of life data were collected, we will apply the utility scores estimated from the other studies according to symptom severity. QALYs will be calculated based on an area under the curve approach. Due to the potentially skewed distribution in costs, a generalised linear mixed model will be employed to analyse costs and QALYs, controlling for baseline patient characteristics, length of follow-up and disease severity. The data will be hierarchical, with individual patients nested within studies, with study modelled as a random effect. Incremental costs and cost-effectiveness and associated CIs will be estimated by bootstrapping.

The economic analyses will follow the same approach as the statistical analyses with calculations undertaken using a one-stage approach.

## Sensitivity analyses

All analyses will be repeated using a two-step approach. The two-stage approach involves calculating the effect for each study separately and then combining the results using traditional meta-analysis techniques. Results using the one-stage and two-stage approach will be compared and any discrepancies explored. This will also allow us to include and explore the impact of the aggregate results from trials for which we are unable to obtain data, as aggregate results can only be included in a two-step approach. Aggregated data from observational studies will not be included because in the absence of full data, it will not be possible to control for confounding by indication.

Sensitivity analysis will explore whether there are differences in inferences based on differences in the approach taken to delayed prescribing and whether there are any differences if studies at high risk of bias are excluded. We will also consider a higher cut-off for presence of fever informed by the literature.

Given the potential long-term problem of antibiotic resistance, a sensitivity analysis will performed to incorporate the costs of antibiotic resistance in the economic analyses. However, given the difficulty previous researchers[31 32] have had in estimating such costs we will vary the assumptions about the likely cost to see whether the inferences are modified.

## Patient and public involvement

The study has two patient and public involvement (PPI) team members who have been actively involved in shaping the research questions and will continue to contribute to all stages of the research. Specifically, they will contribute to data harmonisation, providing advice to ensure the result will yield useful outcomes for patients and members of the public, interpretation of study findings and what they might mean for patients and the public, dissemination at relevant conferences/meetings and how best to disseminate findings to key patient and public groups. In return for their time, PPI representatives will be reimbursed in line with INVOLVE recommendations[33] and will be recognised as co-authors on research outputs.

## Ethics and dissemination

There will be no identifiable patient data in any of the datasets, and data will be stored in a password-restricted folder on the university server. This protocol has been prepared in accordance with the Preferred Reporting Items for Systematic Reviews and Meta-Analyses Protocol guidelines.[34]

The results of this work will provide the best possible evidence regarding when and for whom delayed prescribing is appropriate. The results, disseminated through academic and trade journals, as well as conferences, should help GPs in their decision-making about

prescribing during consultations. The results will also be shared with patients and members of the public through our PPI collaborations. There is potential for PPI to help address the issue of patients seeking antibiotics inappropriately which may help reduce unnecessary prescribing and in turn, antimicrobial resistance.

## DISCUSSION

IPD meta-analysis is more appropriate and has higher statistical power than traditional meta-analysis to identify which key subgroups or patient characteristics may or may not benefit from delayed prescribing. Another advantage of IPD is that some of the biases in studies can be explored more fully than in aggregate data meta-analysis. For example, it is possible to check the randomisation integrity and to perform a full ITT analysis to minimise attrition bias, even if this was not performed for the original publication. Furthermore, with the collaboration and communication required for a successful IPD, it should be possible to obtain information from the original study authors about potential sources of bias. A further advantage of IPD meta-analysis is that it allows study outcomes to be calculated in the same way and the models adjusted for similar confounders which can reduce heterogeneity across studies.

However, there are some challenges associated with IPD meta-analysis.[35] The key risk is not obtaining all the relevant data and failure to do so can result in biased results. It may be particularly difficult to obtain data for older studies, as the data may have been lost or destroyed or the lead researcher may have retired or changed field. We therefore plan to undertake a sensitivity analysis using a two-step approach which will allow us to incorporate the published estimates from any trials for which we are unable to obtain IPD into a meta-analysis. Although IPD offers the ability to conduct additional and more accurate and appropriate analyses, in many cases similar results and conclusions can be drawn from IPD and standard meta-analysis.[35] It is also possible that the outcome measures may not be sufficiently similar to harmonise. An initial scoping review suggests that almost all studies have collected the key outcome measures in a similar way. However, if it is deemed impossible to perform a meta-analysis with the data; a narrative review will be undertaken instead. We hope the results of the study will help GPs to communicate more effectively with patients about the normal course of illness, which patients are likely to benefit from a delayed prescribing approach, the costs of antibiotic use for RTIs and help the general public to feel more confident about waiting to see whether their illness settles before visiting their GP.

**Author affiliations**
[1]Academic Unit of Primary Care and Population Sciences, Faculty of Medicine, University of Southampton, Southampton, UK
[2]Biostatistics Research Group, Department of Health Sciences, College of Life Sciences, University of Leicester, Leiceister, UK
[3]Iberoamerican Cochrane Centre, Department of Clinical Epidemiology and Public Health, Biomedical Research Institute Sant Pau (IIB Sant Pau), Barcelona, Spain
[4]Office for Educational Affairs, Department of General Medicine, University of Rostock, Rostock, Germany
[5]Department of General Practice and Primary Health Care, University of Auckland, Auckland, New Zealand
[6]Southampton Statistical Sciences Research Institute, University of Southampton, Southampton, UK
[7]Divison of Health and Social Care Research, King's College London, London, UK
[8]Basel Institute for Clinical Epidemiology and Biostatistics (CEB), University Hospital Basel and University of Basel, Basel, Switzerland
[9]Institut Català de la Salut, CAP Doctor Carles Ribas, Barcelona, Spain
[10]Division of Population Medicine, School of Medicine, Cardiff University, Cardiff, UK
[11]Centre for Trials Research, College of Biomedical & Life Sciences, Cardiff University, Cardiff, UK
[12]Centre for Academic Primary Care, Population Health Sciences, Bristol Medical School, University of Bristol, Bristol, UK
[13]Institute of General Practice, Rostock University Medical Center, Rostock, Germany
[14]Knowledge and Research Management Unit, Hospital de la Santa Creu i Sant Pau, Barcelona, Spain
[15]Agència de Qualitat i Avaluació Sanitàries de Catalunya (AQuAS), Barcelona, Spain
[16]ASPIRE PPI Panel, Leeds, Institute for Health Sciences, University of Leeds, Leeds, UK

**Acknowledgements** We would like to thank the following collaborators which assisted us by allowing us to use the data from their studies: Dr Jennifer Chao, Dr David McCormick, Ms Jacqueline Nuttall.

**Contributors** BS is the guarantor. BS conceived the original study concept and design and obtained funding from the NIHR Research for Patient Benefit (RfPB) Programme. BS and HH wrote the first draft of the manuscript. GY and SZ contributed to the section on health economics. BS, HH, TB, GY, SZ, PA-C, AA, BA, DB, JB, HCCB, MdlP, NAF, DG, ADH, TK, CL, GM-D, LM, KS, MM and PL contributed to the concept and design of the study and manuscript editing, read, provided feedback and approved the final manuscript.

**Funding** This paper presents independent research funded by the National Institute for Health Research (NIHR) under its Research for Patient Benefit (RfPB) Programme (Grant Reference Number PB-PG-0416-20005). The views expressed are those of the author(s) and not necessarily those of the NHS, the NIHR or the Department of Health & Social Care.

**Disclaimer** The NIHR RfPB is not involved in any other aspect of the project, such as the design of the project's protocol and analysis plan, the collection and analyses. The funder will have no input on the interpretation or publication of the study results.

**Competing interests** None declared.

**Patient consent for publication** Not required.

**Ethics approval** University of Southampton Faculty of Medicine Research Ethics Committee (Reference number: 30068).

**Provenance and peer review** Not commissioned; externally peer reviewed.

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
