## [Reviewer comments · BMJ Open]

ARTICLE DETAILS

TITLE (PROVISIONAL)	Delayed antibiotic prescribing for respiratory tract infections: protocol of an individual patient data meta-analysis
AUTHORS	Stuart, Beth; Hounkpatin, Hilda; Becque, Taeko; Yao, Guiqing; Zhu, Shihua; Alonso-Coello, Pablo; Altiner, Attila; Arroll, Bruce; Böhning, Dankmar; Bostock, Jennifer; Bucher, Heiner; de la Poza, Mariam; Francis, Nick A.; Gillespie, David; Hay, Alastair; Kenealy, Tim; Löffler, Christin; Mas-Dalmau, Gemma; Muñoz, Laura; Samuel, Kirsty; Moore, Michael; Little, Paul

VERSION 1 – REVIEW

REVIEWER	Stephanie Fletcher South Western Sydney Local Health District, Australia
REVIEW RETURNED	19-Oct-2018

GENERAL COMMENTS	This is a very detailed and well written study protocol manuscript. I believe the methodology described is very robust and will add value to the body of knowledge about a very important subject. I have few minor comments:  1. Methods: Is there any exclusion criteria? 2. What is the ISRCTN Registry? 2. It is unclear what the rationale is for using the NHS and PSS perspective for the economic evaluation. Explain and provide more background on what these perspectives entail.
--

REVIEWER	Xiaolin Wei Dalla Lana School of Public Health, University of Toronto, Canada
REVIEW RETURNED	19-Oct-2018

GENERAL COMMENTS	The systematic review is a timely approach to examine the effectiveness of delayed antibiotic prescribing on respiratory tract infections. It is well written and carefully planned. I have the following suggestions:  - The clinical nature of upper and lower respiratory infections is quite different as the former tend to be viral while the latter may be bacterial (or at least some of). Thus, would it make sense to make this difference in your review? - I am confused of how the clinical effectiveness will be estimated. We need a limited number of indicators, preferably less than three, to evaluate the effectiveness. The authors listed all possible indicators which may not agree with each other, for example, patient satisfaction versus re-consultation rates. It will make the review a disarray and inconclusive. I have a feeling that the authors may have too many on the plate to
---

	do such a review. It will be best to focus on a few things, such as the effectiveness, and costs. I am not sure how the authors are going to deal the heterogeneities of the study that includes both RCTs and observational studies. We have done a similar systematic review for RCTs and find it quite problematic: Hu Y, Walley J, Chou R, Tucker JD, Harwell JI, Wu X, Yin J, Zou G, Wei X. Interventions to reduce childhood antibiotic prescribing for upper respiratory infections: systematic review and meta-analysis. J Epidemiol Community Health. 2016;70:1162-70. doi:10.1136/jech-2015-206543
--	--

VERSION 1 – AUTHOR RESPONSE

Reviewer: 1

Reviewer Name: Stephanie Fletcher

Institution and Country: South Western Sydney Local Health District, Australia

Please state any competing interests or state 'None declared': None declared

Please leave your comments for the authors below This is a very detailed and well written study protocol manuscript. I believe the methodology described is very robust and will add value to the body of knowledge about a very important subject. I have few minor comments:

***We thank the reviewer for their encouraging comments on this protocol.

1. Methods: Is there any exclusion criteria?

***Studies on antibiotic prescribing that were not an RCT or observational cohort (for example survey or cross-sectional studies and case-control studies) were excluded. Studies of hospital inpatients were also excluded.

This has been added to lines 144-146 on page 6.

2. What is the ISRCTN Registry?

***The ISRCTN Registry stands for International Standard Randomised Controlled Trial Number (ISRCTN) Registry and is a primary clinical trial registry recognised by WHO and ICMJE that accepts and records all clinical research studies, in order to to improve the publicly available information about clinical studies.

This has been added to lines 153-156 on page 6.

2. It is unclear what the rationale is for using the NHS and PSS perspective for the economic evaluation. Explain and provide more background on what these perspectives entail.

***The National Health Service (NHS) and Personal Social Service (PSS) perspective is the most commonly used approach to economic evaluation conducted in the UK. It covers medication, primary care consultation, walking in centre, NHS telephone service, outpatient attendance, A&E visit and hospital admission. We will take this approach as it makes the most of the data that has been previously reported by trials in this subject area. However, we will also undertake a further analysis from the societal perspective, including the societal cost of antimicrobial resistance and time off work

due to illness as a sensitivity analyses.
This has been added to page 7, lines 286-297.

Reviewer: 2
Reviewer Name: Xiaolin Wei

Institution and Country: Dalla Lana School of Public Health, University of Toronto, Canada

Please state any competing interests or state 'None declared': None declared.

Please leave your comments for the authors below The systematic review is a timely approach to examine the effectiveness of delayed antibiotic prescribing on respiratory tract infections. It is well written and carefully planned. I have the following suggestions:

***We thank the reviewer for their interest in this protocol and helpful comments.

- The clinical nature of upper and lower respiratory infections is quite different as the former tend to be viral while the latter may be bacterial (or at least some of). Thus, would it make sense to make this difference in your review?

***The reviewer is right. We will assess the association of antibiotic prescribing approach with outcomes of interest (symptom severity, duration of illness, re-consultations, etc.) using data from all studies and including all types of infections, but will also repeat the analyses according to diagnostic group (acute sore throat, cough/chest infection, otalgia/otitis media, upper respiratory tract infection (URTI), separately.

We have specified this in lines 253-256 on page 8.

- I am confused of how the clinical effectiveness will be estimated. We need a limited number of indicators, preferably less than three, to evaluate the effectiveness. The authors listed all possible indicators which may not agree with each other, for example, patient satisfaction versus re-consultation rates. It will make the review a disarray and inconclusive.

***The study has a primary outcome – symptom severity – which will be our main indicator of clinical effectiveness. However, we will also assess the clinical effectiveness for each of our secondary outcomes, duration of illness, development of complications and re-consultation. We will then take a narrative approach and describe the findings for each indicator separately. We agree that the “clinical effectiveness” for patient satisfaction may not be consistent with the other indicators and should this be the case, we will reflect on this in the discussion and recommendations.

We have added this to lines 262-266 on page 8.

I have a feeling that the authors may have too many on the plate to do such a review. It will be best to focus on a few things, such as the effectiveness, and costs.

***We thank the reviewer for this comment. We will focus primarily on clinical effectiveness of delayed antibiotic prescribing on symptom severity as this is our primary outcome, assess the key subgroups for whom delayed antibiotic prescribing is beneficial, and determine the cost effectiveness of the delayed antibiotic approach. We will assess clinical effectiveness of delayed prescribing for duration of illness, development of complications, re-consultation and patient satisfaction as secondary outcomes as these were included in our funded application and our PPI work has suggested that patients find these outcomes important.

I am not sure how the authors are going to deal the heterogeneities of the study that includes both RCTs and observational studies. We have done a similar systematic review for RCTs and find it quite problematic:

Hu Y, Walley J, Chou R, Tucker JD, Harwell JI, Wu X, Yin J, Zou G, Wei X. Interventions to reduce childhood antibiotic prescribing for upper respiratory infections: systematic review and meta-analysis. *J Epidemiol Community Health*. 2016;70:1162-70. doi:10.1136/jech-2015-206543

***Thank you for this helpful paper. We agree that the synthesis of different study designs can be challenging and, should it prove impossible due to very high levels of heterogeneity, we will present the findings from observational and RCT studies separately. However, work on this sort of cross design synthesis by the Program and Methodology Division of the US General Accounting Office (1992) has provided a theoretical framework for combining database data and RCT data. Drawing on this work, the current review will apply a similar approach to the synthesis of observational cohort and RCT data. Propensity score analysis, a technique developed to help balance observational studies on key covariates, can help to adjust for confounding by indication on measured covariates. This serves to balance the groups and make them more like those in a randomised control trial. Observational studies, after adjustment for potential confounding, will be included in the main analysis if appropriate. Sources of heterogeneity will be explored and explained and a random effects model will be used for all analyses. Even if it is possible to pool these with the RCTs in a meta-analysis as part of the evidence synthesis process, the results of observational studies will also be reported separately to those of the RCTs as a sensitivity analysis. We have added this to lines 268-282 on page 8.

Reference: General Accounting Office. Cross-design synthesis: A new strategy for medical effectiveness research. March 1992. Washington DC